# Advanced Maternal Age Deteriorates the Developmental Competence of Vitrified Oocytes in Mice

**DOI:** 10.3390/cells10061563

**Published:** 2021-06-21

**Authors:** Ju Hee Lee, Jae Kyun Park, Sook Young Yoon, Eun A Park, Jin Hyun Jun, Hyunjung J. Lim, Jayeon Kim, Haengseok Song

**Affiliations:** 1Department of Biomedical Sciences, CHA University, Seongnam 13488, Korea; joohee8406@naver.com (J.H.L.); pjk987427@chamc.co.kr (J.K.P.); 2Fertility Center of Gangnam CHA Medical Center, CHA University, Seoul 06125, Korea; syyoon11@cha.ac.kr; 3CHA Fertility Center Seoul Station, CHA University, Seoul 04637, Korea; eapark9327@chamc.co.kr; 4Department of Biomedical Laboratory Science, College of Health Science, Eulji University, Seongnam 13135, Korea; junjh55@hanmail.net; 5Department of Veterinary Medicine, School of Veterinary Medicine, Konkuk University, Seoul 05029, Korea; hlim@konkuk.ac.kr

**Keywords:** advanced maternal age (AMA), oocyte vitrification, oocyte quality, developmental competence, fertility preservation, time-lapse monitoring

## Abstract

Advanced maternal age (AMA) is known to be related to the decrease in the quality and quantity of oocytes. Oocyte vitrification is now considered an established assisted reproductive technology for fertility preservation. However, it remains unclear whether the oocytes in older women are more sensitive to various insults during vitrification. Thus, we evaluated whether AMA affects cellular and molecular features and developmental outcomes of oocytes after vitrification in mice. The oocytes were grouped as young fresh (YF), young vitrified/warmed (YV), aged fresh (AF), and aged vitrified/warmed (AV). The survival rate of AV oocytes was significantly lower than that of YV oocytes. The rates of fertilization, cleavage, and blastocyst formation of AV oocytes were significantly lower than those of other groups. AV oocytes were represented as aberrations in mitochondria distribution, microvacuole size, and autophagosome formation, leading to delayed embryo development in mice. This delay was associated with a reduced number of total cells and trophectoderm in the blastocyst developed from AV oocytes. Collectively, AMA exaggerates the vulnerability of oocytes to cryo-damage that occurs during vitrification in mice, suggesting that the current vitrification protocols optimized for oocytes from young females should be modified for oocytes from aged women.

## 1. Introduction

Over the past 10 years, advances in oocyte vitrification techniques have led to a significant change in the reproductive choices of women [1]. The reproductive ability of women begins to decline significantly at the age of 35 and more rapidly after 40 years of age [2]. Advanced maternal age (AMA) is associated with a decline in both ovarian reserve and oocyte competence [3]. This reproductive aging reduces the embryo implantation potential as well [4]. Single women who do not want to conceive can postpone childbearing by cryopreserving oocytes. It was reported that an increasing number of women pursue oocyte cryopreservation [5] and the median age of these women is 38 years [6,7]. Of note, more than half the women who undergo oocyte cryopreservation are older than 37 years.

Vitrification has been widely used as a new cryopreservation technique using an ice-free cryopreservation procedure [8]. Mammalian oocytes have a relatively large size (a diameter of 90 and 125 µm for mice and humans, respectively), which makes the genetic material inside cells more susceptible to insults, such as temperature changes and chemical reagents. Vitrification has adverse effects on the oocytes, such as spindle confusion, zona pellucida hardening, and death, mainly because of the exposure to high concentrations of cryoprotectant agents (CPAs) that have toxic effects on cells [9]. To minimize possible damages to the oocytes during vitrification, the procedures have been extensively modified regarding the combination, concentration, composition, and exposure time of CPAs [10]. In addition, a super-rapid cooling and/or warming using slush nitrogen improved the clinical outcomes of oocyte vitrification [11]. Owing to the advances in vitrification techniques, the clinical outcomes of in vitro fertilization (IVF) using cryopreserved oocytes are now comparable to those of using fresh oocytes [12]. However, most clinical studies about oocyte cryopreservation investigated donor oocytes from young women. Although there is limited data showing the clinical outcomes of oocyte cryopreservation of older women (aged 37 or older), those data showed very poor clinical outcomes that are apparently different from those in young women [13,14].

The developmental competence of oocytes tends to be reduced as they age [15,16]. Oocytes from aged females in both humans and mice have a high incidence of anomalies in the meiotic spindle and chromosome alignment [17]. A previous study found that fresh oocytes from aged mammals had mitochondrial activity dysfunction [18], which is associated with spindle formation, chromosomal segregation, and fertilization, resulting in decreased quality of oocyte and aneuploidy [19]. The aging process also negatively influenced the quality of embryos developed from vitrified oocytes. The pronucleus formation and first cleavage of vitrified oocytes were significantly decreased in aged mice [20,21]. Taken together, it can be hypothesized that the cellular vulnerability of aged oocytes to cryo-damage may contribute to the poor clinical outcomes of oocyte vitrification in older women. Considering the increasing need of oocyte cryopreservation from old women and the cellular vulnerability of oocytes to damage caused by vitrification/warming in aged women, the optimization of the vitrification protocol for oocytes from aged individuals as a personalized medical treatment is warranted. However, it remains unclear how the oocyte aging in vivo affects the quality of oocytes for their survival after vitrification and the following developmental competence. Thus, the purpose of this study was to understand whether AMA aggravates the detrimental effects of vitrification on the quality of oocytes at the cellular and molecular level.

## 2. Materials and Methods

### 2.1. Animals

All experimental protocols were performed in accordance with guidelines of the Declaration of Helsinki. All animals used in this study were maintained and handled according to the policies approved by CHA University Institutional Animal Care and Use Committee (approval number 190201). C57BL/6 (BL6) female mice at various ages were provided by Orient Bio (Gapyeong, Gyeonggi, Korea) or JA BIO (Suwon, Gyeonggi, Korea). Mice (1- to 2-month-old and 10- to 18-month-old) were considered young and aged, respectively. Orient Bio provided 2-month-old BDF1 male mice for IVF. All mice were housed under temperature- and light-controlled conditions (12 h-light-dark cycle) and fed *ad libitum*.

### 2.2. Oocyte Collection

Mice were injected intraperitoneally with 10 IU of pregnant mare serum gonadotropin (PMSG; Daesung, Uiwang, Gyeonggi, Korea), followed by 10 IU human chorionic gonadotropin (hCG; Daesung) at 48 h post-PMSG. Ovulated cumulus-oocyte complexes (COCs) were retrieved from the oviduct at 13 h post-hCG. To remove the cumulus cells, COCs were treated with 0.1% hyaluronidase (Sigma-Aldrich, St. Louis, MO, USA) at room temperature. Denuded oocytes were washed two times with M2 medium (Sigma-Aldrich) containing 1% penicillin-streptomycin (Hyclone, Logan, UT, USA) and collected in EmbryoMax^®^ KSOM + AA with D-glucose and Phenol Red (KSOM; MR-121-D, Millipore, Danvers, MA, USA). Oocytes were classified as normal or abnormal using a stereomicroscope according to the morphological differences in the ooplasm. Oocytes were considered abnormal when they presented cytoplasm compaction, a large perivitelline space (PVS), and debris in the PVS.

### 2.3. Vitrification and Warming

Ethylene glycol (EG, Sigma-Aldrich) and dimethyl sulfoxide (DMSO, Sigma-Aldrich) were used as CPAs in a vitrification solution. Oocytes were equilibrated in Quinn’s Advantage Medium with HEPES (HEPES; ART-1023, SAGE Media, Trumbull, CT, USA)-based media containing 7.5% EG, 7.5% DMSO, and 20% (*v*/*v*) human serum albumin (HSA; ART-3001, SAGE Media) for 2.5 min, and then transferred to media containing 15% EG, 15% DMSO, and 0.5 M sucrose (Sigma-Aldrich) for 30 s. Equilibrated oocytes were loaded onto a copper grid (Ted Pella Inc. Redding, CA, USA) and directly dipped in liquid nitrogen (LN_2_). Vitrified oocytes were kept in the LN_2_ for 10 min and warmed for the experiments. For the warming procedure, the grid was removed from LN_2_ and directly placed in the warming solution (0.5 M sucrose and 20% HSA in HEPES) for 2.5 min. Warmed oocytes were sequentially incubated in descending concentrations of sucrose (0.25, 0.125, and 0 M) for 2.5 min each. Vitrified/warmed (VW) oocytes were washed two times with M2 media. Finally, VW oocytes were cultured in KSOM medium and incubated at 37 °C in 5% CO_2_ for 2 h for recovery. Most oocytes appeared recovered without noticeable morphological deformation, such as cytoplasmic lysis or fragmentation. The survived oocytes were then collected and used for further experiments. The rate of survival (number of survived oocytes/VW oocytes) was evaluated and compared between experimental groups.

### 2.4. Immunofluorescence Staining of Oocytes and Blastocysts

For oocyte immunofluorescence staining, oocytes were fixed in 4% paraformaldehyde containing 0.1% polyvinyl alcohol (PVA) in phosphate-buffered saline (PBS) for 1 h at 4 °C. Fixed oocytes were washed three times with PBS for 10 min each, transferred onto a glass-slide, and then treated with 0.2% Triton X-100 in PBS for 10 min for permeabilization. Nonspecific staining was blocked using Protein Block Serum-Free (Dako, Carpinteria, CA, USA) for 1 h, followed by incubation with the primary antibodies α-tubulin (1:100, ab15246, Abcam, Cambridge, UK), and pericentrin (PCNT; 1:100, 611814, BD Biosciences, San Jose, CA, USA) overnight at 4 °C. Oocytes were labeled with Alexa Fluor^®^ 488 and Alexa Fluor^®^ 594 conjugated secondary antibodies (1:1000, Invitrogen, Carlsbad, CA, USA) for 1 h at room temperature. The nuclei were stained with DAPI (1:1000, 62248, Thermo Fisher Scientific, Fair Lawn, NJ, USA). Oocytes were mounted using Gel Mount Aqueous Mounting Medium (M01, Sigma-Aldrich) and observed using a confocal microscope (Zeiss LSM880, Carl Zeiss, Oberkochen, Germany).

For blastocyst staining, blastocysts were fixed 6 days after IVF, and permeabilization and blocking were performed sequentially. Blastocysts were labeled with a primary OCT4 antibody (1:100, 611203, BD Biosciences), and Alexa Fluor^®^ 594 conjugated secondary antibody (1:1000, Invitrogen), followed by counterstaining with DAPI to label the nuclei. The number of trophectoderm (TE) cells was calculated as the total cell number minus the inner cell mass (ICM) cell number.

### 2.5. Oocyte Live Imaging Using a Confocal Microscope

To observe the mitochondrial distribution in oocytes before and after vitrification, live oocytes were washed two times with M2 media and stained with MitoTracker^TM^ Red CMXRos (200 nmol/L, M7512, Invitrogen) at 37 °C in 5% CO_2_ for 15 min. They were then treated with NucBlue^TM^ Live Cell Stain ReadyProbes^TM^ reagent (R37605, Invitrogen) for 15 min in an incubator. Oocytes were washed two times with M2 media and transferred to a glass-bottom confocal dish (P35G-1.0-14-C, MatTek, Ashland, MA, USA). Live images of mitochondria in oocytes (red color) were obtained directly using a confocal microscope (Zeiss LSM880), and analyzed using ZEN software (ZEN 2012, Blue edition, Carl Zeiss) after red was converted to pseudo-color (green) for better visualization. For quantification of the mitochondrial distribution, fluorescence intensity profiles were measured using the ImageJ software (v1.48, NIH, Bethesda, Maryland, USA).

### 2.6. Ultrastructural Observation of Oocytes Using Transmission Electron Microscopy (TEM)

Oocytes were first fixed in 2% glutaraldehyde (Sigma-Aldrich) in PBS for 3 h at room temperature and briefly washed three times with 0.1% PVA in PBS. Oocytes were placed in agar chips for pre-embedding and post-fixed in 1% osmium tetroxide (Sigma-Aldrich) in PBS. After dehydration and infiltration, the embryos were embedded in Epon 812. Thin sections were cut using a diamond knife in an Ultracut UCT Ultramicrotome (Leica, Wetzlar, Germany) and placed on copper grids. The sections were double-stained with 2% uranyl acetate and lead citrate, and examined using an H7600 transmission electron microscope (80 kV; Hitachi, Tokyo, Japan). Representative semi-thin and ultra-thin sections of each oocyte were selected and evaluated for the number and diameter of microvacuoles, respectively by two independent observers.

### 2.7. Quantitative Analysis of Autophagosome Formation in Oocytes

To observe the formation of autophagosomes in VW oocytes from young and aged mice, oocytes were stained with CYTO-ID Green dye (1:500, Enz-51031-0050, Enzo life sciences, Farmingdale, NY, USA) and Hoechst (1:1000, H33258, Sigma-Aldrich) at 37 °C in 5% CO_2_ for 30 min. CYTO-ID Green dye, a cationic amphiphilic tracer dye, specifically labels autophagic compartment, indicating that the puncta stained with CYTO-ID are a specific autophagy marker [22]. The oocytes from young mice were incubated with 100 nM rapamycin (Enz-51031-0050, Enzo life sciences) at 37 °C in 5% CO_2_ for 2 h as the control of autophagosome formation in oocytes before CYTO-ID Green staining. The oocytes were fixed, permeabilized, and then incubated with Phalloidin (1:100, A22287, Invitrogen) in PBS for 15 min. Oocytes were washed three times with PBS for 10 min each. After a final wash in PBS, the oocytes were directly placed onto a glass-bottom confocal dish (P35G-1.0-14-C, MatTek), observed using a confocal microscope (Zeiss LSM880, Carl Zeiss), and analyzed using ZEN software (Carl Zeiss).

### 2.8. IVF

Sperm was squeezed out of the cauda epididymis, which was dissected from 2-month-old BDF1 male mice. The retrieved sperm was transferred immediately into a pre-warmed drop of Human Tubal Fluid media (HTF; FUJIFILM Irvine Scientific, San Diego, CA, USA) with 10% KnockOut^TM^ serum replacement (Thermo Fisher Scientific) for sperm capacitation for 1 h. Sperms and oocytes were then inseminated in HTF media for 6 h. Finally, the zygotes were cultured in KSOM medium. The rates of fertilization (number of 2-cell embryos/survived oocytes), cleavage (number of 4-cell embryos/survived oocytes), and blastocyst formation (number of blastocysts/survived oocytes) were evaluated between experimental groups.

### 2.9. Time-Lapse Monitoring of Embryo Morphokinetics

Images of each embryo were acquired every 10 min for the whole duration of culture and merged to provide time-lapse information of embryo morphokinetics. The exact time of developmental and morphological events was measured from the time of sperm addition into oocyte-containing HTF media (insemination, t0). Fertilized zygotes with two pronuclei were transferred to a Primo Vision culture dish (16606, Micro well group culture dish, Vitrolife, Gothenburg, Sweden) containing KSOM medium. All embryos in the culture dish were monitored using a Primo Vision Time-Lapse Microscope (Vitrolife) until day 6 of embryo development. The morphokinetics of embryos were recorded to demonstrate the precise timing of the cell divisions in correlation to the timing of insemination. The cleavage time from t0 to 2-cell (t2), 3-cell (t3), 4-cell (t4), 6-cell (t6), 8-cell (t8), morula (tM, the time for compaction), and blastocoel formation (tEB) was expressed as hours post insemination (HPI).

### 2.10. Statistics

Statistical tests were performed using GraphPad Prism v8 software (GraphPad Software, La Jolla, CA, USA). Analyses were performed using one-way ANOVA followed by *post hoc* analysis, chi-squared test, or Fisher’s exact test. Values represent the mean ± standard error (SEM) or median. A *p*-value less than 0.05 was considered statistically significant. * *p* < 0.05, ** *p* < 0.01, *** *p* < 0.005.

## 3. Results

### 3.1. AMA Decreases the Total Number of Ovulated Oocytes and Accelerates Morphologic Abnormalities in Oocytes

The age-wise comparison between humans and mice based on the life span was illustrated to define AMA in mice (Figure 1A). As shown in Figure 1B, the total number of ovulated oocytes per mouse was inversely correlated with age. The number of ovulated oocytes from 1- (28.3 ± 1.9) to 6-month-old (23.3 ± 1.7) mice was similar, but it sharply started to decline in 8-month-old (12.7 ± 0.7) mice, reaching down to 8.0 ± 0.7 and 1.6 ± 0.4 in 12- and 18-month-old mice, respectively. The number of morphologically abnormal oocytes, defined with compaction of the ooplasm and/or large PVS, increased in an age-dependent manner (8.8% in 1-month-old and 35.3% in 10-month-old mice, Figure 1C,D). The survival rates of VW oocytes from young (1- to 2-month-old) and aged (10- to 12-month-old) mice were 94.2% (308/327) and 74.4% (93/125) (*p* < 0.005), respectively (Figure 1E and F). In the following experiments, only morphologically normal oocytes were used.

### 3.2. AMA, but Not Vitrification, Deteriorates the Spindle Structures of Metaphase Chromosomes in Oocytes

To evaluate the effects of vitrification and AMA on meiotic spindles in oocytes, immunofluorescence staining for α-tubulin in microtubules (a spindle marker) and PCNT (a spindle pole marker) was performed (Figure 2A). Fresh oocytes from young mice (YF oocytes) showed a representative organization of spindles (α-tubulin as green and PCNT as red) with meiotic chromosomes (blue). Microtubules were organized in a barrel shape in the metaphase spindle. Although the spindle structures were not disturbed in VW oocytes from young mice (YV oocytes), abnormal spindles were often observed in both fresh and VW oocytes from aged mice (AF and AV oocytes, respectively) (Figure 2A and B). For example, spindles were frequently organized into misaligned chromosomes (arrow) and decondensed spindle poles (arrowhead) (Figure 2A).

### 3.3. AMA and Vitrification Cause Abnormal Mitochondrial Aggregation in the Central Area of Oocytes

In YF oocytes, mitochondria showed a typical even distribution throughout the ooplasm. However, most YV oocytes (54/58, 93.1%) had abnormally aggregated mitochondria in the central area (Figure 3A,B). AMA alone disturbed the even distribution of mitochondria in 78.6% of AF oocytes, and after vitrification, all AV oocytes had aggregated mitochondria in the central area (100%). From the intensity profiles, it was observed that YF oocytes had an average intensity of 31.98 arbitrary units (A.U.) of mitochondria within 10 µm from the plasma membrane, whereas YV, AF, and AV oocytes had an average intensity of 8.88, 19.66, and 9.57 A.U., respectively (Figure 3C). These results suggest that AMA, as well as vitrification, disturb the even distribution of mitochondria in oocytes.

### 3.4. AMA Significantly Increased the Size of the Microvacuoles in VW Oocytes

TEM images of oocytes (Figure 4A) showed that the number of microvacuoles was relatively increased in YV and AV oocytes compared to fresh oocytes, although the difference was not statistically significant (Figure 4B). The median sizes of microvacuoles were similar among YF, YV, and AF oocytes (477, 560, and 454 nm, respectively). However, the size of microvacuoles in AV oocytes (980 nm) significantly increased compared to all the other groups (Figure 4C). Furthermore, 34.5% (38/110) of microvacuoles in AV oocytes were larger than 1000 nm.

In YF oocytes treated with 100 nM rapamycin as a control for autophagy formation, many puncta stained with CYTO-ID were observed (shown as green dots). When the number of puncta was counted, 50% of AV oocytes (7/14) had more than five puncta (Figure 4D,E), although YF, YV, and AF oocytes had none or few. When the autophagy patterns in oocytes were classified based on the number of puncta, only AV oocytes showed a similar pattern to YF oocytes treated with rapamycin (Figure 4F).

### 3.5. AMA Deteriorates the Developmental Potential of VW Oocytes

The fertilization rates were comparable among YF, YV, and AF oocytes (92.4%, 89%, and 86.1%, respectively). However, the fertilization rate of AV oocytes was significantly reduced (42/64, 65.6%), although morphologically abnormal oocytes were excluded before IVF (Figure 5A), suggesting that AMA itself deteriorates the fertilization potential of oocytes after vitrification. Furthermore, the cleavage rates were significantly lower in AV oocytes (59.4%) than in the other three groups (90.5%, 87.7%, and 83.5%, respectively) (Figure 5B). The blastocyst formation rate of AF oocytes (63.3%) was significantly lower than that of YF oocytes (81.9%). Vitrification further significantly reduced the blastocyst formation rate in AV oocytes (40.6%) (Figure 5C).

### 3.6. Embryo Morphokinetics of AV Oocytes Were Significantly Retarded

To examine whether AMA affects the morphokinetics of embryos derived from VW oocytes, each blastomere cleavage, compaction, and blastocoel formation was evaluated using time-lapse monitoring systems. The cleavage time of embryos from AV oocytes was longer throughout embryo development from t2 to tEB (Figure 6A,B). The cleavage time of t2 in AF (20.4 HPI) and AV (20.4 HPI) oocytes was longer compared to that in YF (19.2 HPI) and YV (19.3 HPI) oocytes (Figure 6C). From t4 to tEB, embryos from AF oocytes showed comparable morphokinetics to YV and YF oocytes (Figure 6D–H). However, early development was persistently delayed in embryos from AV oocytes. Especially, the cleavage time of embryos from AV oocytes at t8, tM, and tEB (60.0, 68.9, and 98.3 HPI, respectively) was significantly longer than that of embryos from AF oocytes (55.9, 65.7, and 85.0 HPI, respectively). These results suggest that AMA adversely affects not only the blastocyst formation rate, but also the morphokinetics of embryos from VW oocytes.

### 3.7. AMA Was Associated with a Significant Reduction in the Cell Number of the Blastocysts from VW Oocytes

Immunostaining of OCT4 (ICM marker) with DAPI in blastocysts developed from oocytes in all four groups showed that most blastocysts from YF, YV, and AF oocytes appeared to be similar to each other (Figure 7A). However, blastocysts from AV oocytes had a significantly lower number of cells compared to the blastocysts from the other groups (Figure 7B). Furthermore, the number of TE in the blastocysts from AV oocytes was significantly reduced compared to that of YV and AF oocytes (Figure 7C). The ICM/TE ratio was comparable among all four groups since not only the number of TE, but also the ICM in AV oocytes, was reduced (Figure 7D,E).

## 4. Discussion

As an increasing number of old women pursue oocyte vitrification [24], the importance of the optimization of oocyte vitrification for aged women has been emphasized. We clearly demonstrated evidence that AMA demands further modification of oocyte vitrification to accomplish successful pregnancy outcomes regarding multiple aspects. It is strongly suggested that the current protocols optimized for preserving oocytes from young women should be modified for oocytes from aged women as personalized medicine. Furthermore, we showed that mice recapitulate the age-dependent phenotype of ovulated oocytes seen in humans. Mouse oocytes exhibited an age-dependent increase in morphologic abnormalities, which negatively influence the oocyte survival rate and subsequent embryo development. In addition, aging is correlated with a higher incidence of spindle defects and chromosomal misalignments in mouse and human oocytes [25]. While the meiotic spindle is one of the most sensitive cytoplasmic organelles during vitrification [26], we showed that vitrification does not cause any further defects in chromosomal segregation and spindle assembly in oocytes from aged mice. Aberrant spindle assembly in oocytes is known to be associated with chromosomal loss and aneuploidy of the resulting embryos [19]. Considering that the aneuploidy rate of embryos is increased in an age-dependent manner, our results suggest that AMA, not vitrification-induced cryo-damage, may be a significant factor associated with the increased number of chromosomal errors in AV oocytes.

The mitochondrial distribution pattern in oocytes is linked to developmental competence in several species, including mice [27], cows, and pigs [28]. Mitochondrial aggregation is an event upstream of cytochrome *c* release during apoptosis [29]. In this study, mitochondria were evenly distributed throughout the cytoplasm of YF oocytes. However, mitochondria aggregated into the central area, not only in YV and AV oocytes, but also in AF oocytes, suggesting the negative impact of AMA on mitochondria dynamics in mouse oocytes. Exposure to CPAs during vitrification and warming causes osmotic dehydration and shrinkage in oocytes. Such changes may cause a redistribution of mitochondria in oocytes [30] and affect the clinical outcome of assisted reproductive technology (ART) using VW oocytes. Since mitochondria localize in areas where high levels of ATP are required, alterations in mitochondria distribution may be crucial for oocyte function and metabolism [31]. An even distribution of mitochondria is believed to be important during cleavage to ensure that each blastomere receives sufficient mitochondria to survive. Thus, AMA and vitrification could induce mitochondrial aggregation in oocytes that may lead to adverse embryo development after fertilization. This was supported by our data that showed lower rates of fertilization and blastocyst formation in AV oocytes. These results are consistent with human data that showed lower survival and pregnancy rates of IVF cycles using VW oocytes from older women [32].

In this study, ultrastructural cytoplasmic alterations in AV oocytes suggest a greater sensitivity to stress response related to vitrification. In humans, microvacuolization was observed as a hallmark of cell damage after slow freezing of oocytes from young women [33]. Cytoplasmic microvacuoles are usually rare or absent in fresh, healthy MII oocytes in humans [34]. It was reported that vitrification causes microvacuolization in bovine [35] and horse oocytes [36]. It is consistent with our results showing that VW oocytes from both young and aged mice had a relatively higher number of microvacuoles than their fresh counterparts. Microvacuoles may be formed by the swelling of oocytes due to osmotic stress during vitrification and warming [34,37]. In this aspect, it should be noted that many microvacuoles in the periphery of AV oocytes were significantly larger than those in oocytes of other groups. These larger microvacuoles may reflect the vulnerability of oocytes to osmotic stress that occur only in AV oocytes during vitrification and warming.

It is also possible that these large microvacuoles in AV oocytes were newly formed as a result of the cytoplasmic damage [36]. In fact, microvacuoles are known to be related to autophagic vesicle formation [38]. MII oocytes showed a low level of LC3 positive puncta (autophagy) until fertilization [39], but autophagic activation is observed in oocytes exposed to any stress, such as vitrification [40]. Thus, larger microvacuoles in AV oocytes may be associated with an increased autophagic process. Of note, AV oocytes, but not the others, showed a similar autophagosome formation in oocytes treated with rapamycin, a widely used inducer of autophagy. It suggests that the increased autophagic response in AV oocytes may lead to lower rates of fertilization and blastocyst formation. Further studies will be warranted to examine whether the suppression of induced autophagy in AV oocytes is beneficial for fertilization and subsequent embryo development.

Time-lapse monitoring of preimplantation embryos, in conjunction with traditional embryo scoring based on morphology, is an important tool to evaluate embryo quality for making critical decisions in ART [41,42]. In humans, developmental timing is altered, and blastocyst formation is delayed in embryos derived from VW oocytes from young women, compared with their fresh counterparts [43]. In this study, while vitrification did not disturb the developmental competence of oocytes from young mice, it cumulatively deferred the preimplantation development of oocytes from aged mice. Furthermore, wide ranges of morphokinetics of embryos from AV oocytes could partly explain the poor outcomes obtained using the AV oocytes. It is interesting to examine whether the embryos from AV oocytes after embryo transfer are postnatally as healthy as ones from other groups. In addition, AMA reduces the number of total cells, ICM, and TE, even though some AV oocytes could develop into the blastocyst stage. A positive correlation between the total cell number of the blastocysts and the subsequent embryo development after implantation was reported [44,45]. The trophectoderm morphology grade, based on the cell number and organization, was significantly correlated with implantation and ART outcomes [46]. These findings suggest that AMA further deteriorates the vulnerable status of VW oocytes, leading to retardation of preimplantation embryo. This could be associated with aberrant expression of genes that are involved in oxidative stress, mitochondria function, and chromosome segregation in aged oocytes [47,48]. While this study has a limitation that the findings may not be directly translated to human oocytes since all the data came from mouse oocytes, it strongly suggests that the vitrification protocols should be modified and/or improved for oocytes from women at an AMA. Additional studies are also needed in other mouse strains and species, to confirm the general validity of AMA effects observed in this work.

## 5. Conclusions

The present data clearly demonstrate that an AMA further aggravates the vulnerability of the VW oocytes to cryo-damage during vitrification/warming at the cellular and molecular level, leading to delayed and poor preimplantation embryo development in mice. While this study was performed in mouse oocytes, mice recapitulate an age-dependent phenotype of ovulated oocytes of humans. Accordingly, we suggest that the current vitrification protocols optimized for oocyte from young donors need to be modified and/or improved to efficiently preserve the quality of oocytes from women at an AMA.

## Figures and Tables

**Figure 1 cells-10-01563-f001:**
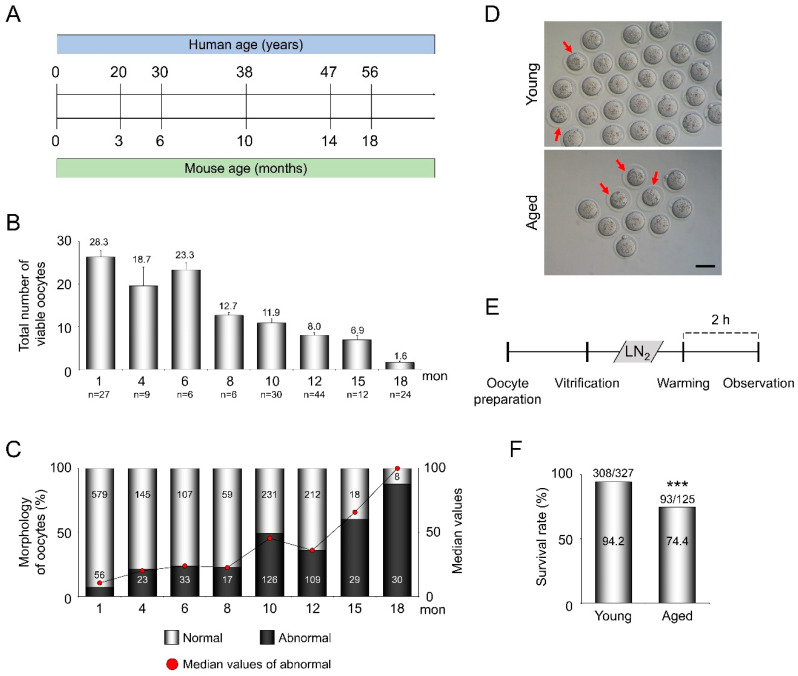
Age-dependent changes in the number and morphological abnormalities of ovulated oocytes. (**A**) A schematic comparison of the life span between humans and C57BL/6 (BL6) mice, which has been modified from “Mouse models in aging research” [23]. (**B**) The total number of ovulated oocytes from BL6 mice by age (from 1- to 18-month-old). The graph presents the average number of ovulated oocytes per mouse (*n* = 6–44 mice in each group). (**C**) An age-dependent increase in morphologically abnormal oocytes (1- to 18-month-old). White and black bars represent the percentage of normal and abnormal oocytes, respectively (*n* = 38–635 oocytes in each group). Red dots represent the median values in each group. (**D**) Representative microscopic images of ovulated oocytes from young and aged BL6 mice after superovulation. Red arrowheads indicate morphologically abnormal oocytes. (**E**) A schematic diagram to illustrate the experimental procedures to evaluate whether the survival rates of oocytes from young and aged mice are different between strains after vitrification. (**F**) Comparison of the survival rates of VW oocytes from young (1- to 2-month-old) and aged (10- to 12-month-old) mice. The graphs present the survival rate of total ovulated VW oocytes (*n* = 125–327 oocytes in each group). Scale bar = 75 µm. Statistical comparisons were performed using chi-squared test. *** *p* < 0.005.

**Figure 2 cells-10-01563-f002:**
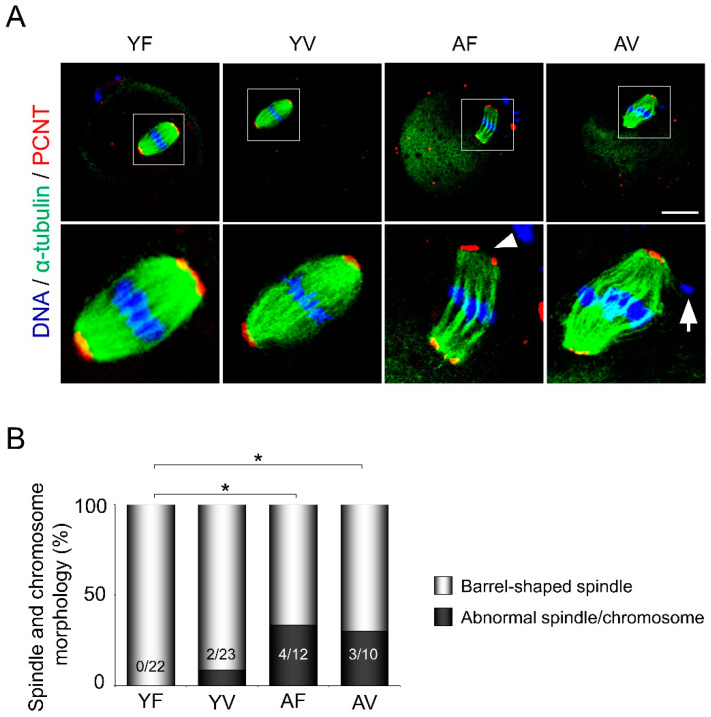
Abnormal spindle and chromosome organization in oocytes from aged mice. (**A**) Representative images of immunofluorescence staining for α-tubulin (a spindle marker, Green), pericentrin (PCNT; a spindle pole marker, Red), and DAPI (DNA, Blue) in oocytes from young and aged mice with or without vitrification. The bottom panels show the enlarged images of the boxed areas in the upper panels. Arrowhead and arrow indicate abnormal spindle/spindle pole and chromosome, respectively. Note that abnormal spindle and chromosome organization were often observed in oocytes from aged mice. (**B**) Quantitative analyses of oocytes with abnormal spindle and chromosomal morphologies (*n* = 10–23 oocytes in each group). White and black bars represent the percentage of oocytes with normal and abnormal morphologies, respectively Scale bar = 20 µm. Statistical analyses were performed using Fisher’s exact test. * *p* < 0.05. YF: Young fresh, YV: Young vitrified, AF: Aged fresh, AV: Aged vitrified.

**Figure 3 cells-10-01563-f003:**
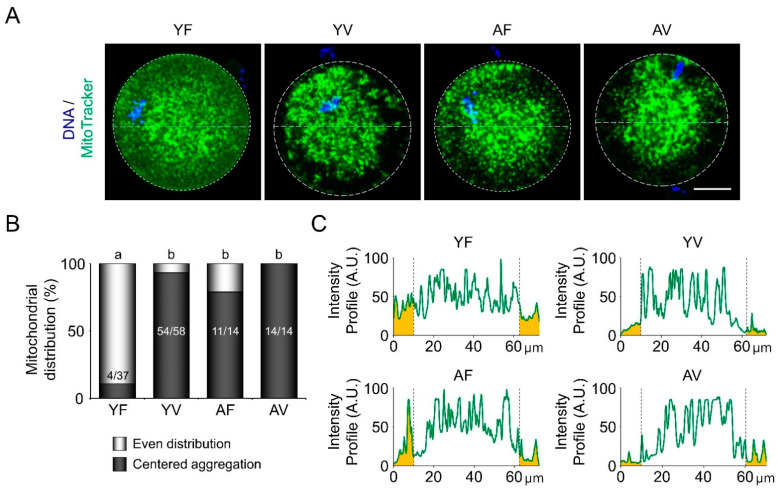
Fluorescence live cell imaging to examine the patterns of mitochondrial distribution after vitrification in oocytes. (**A**) Representative microscopic images of mitochondrial distribution visualized using MitoTracker dye (Green) in live oocytes from young and aged mice before or after vitrification. (**B**) Quantitative analyses of mitochondrial distribution patterns (*n* = 14–58 oocytes in each group). Mitochondrial distribution was divided into even distribution and centered aggregation. White and black bars represent the percentage of even distribution and centered aggregation, respectively. (**C**) The graphs present the mitochondrial intensity profiles throughout the ooplasm of oocytes. Scale bar = 20 µm. Statistical analyses were performed using Fisher’s exact test. Different superscript letters (a–b) denote a significant difference from the control, *p* < 0.05. YF: Young fresh, YV: Young vitrified, AF: Aged fresh, AV: Aged vitrified.

**Figure 4 cells-10-01563-f004:**
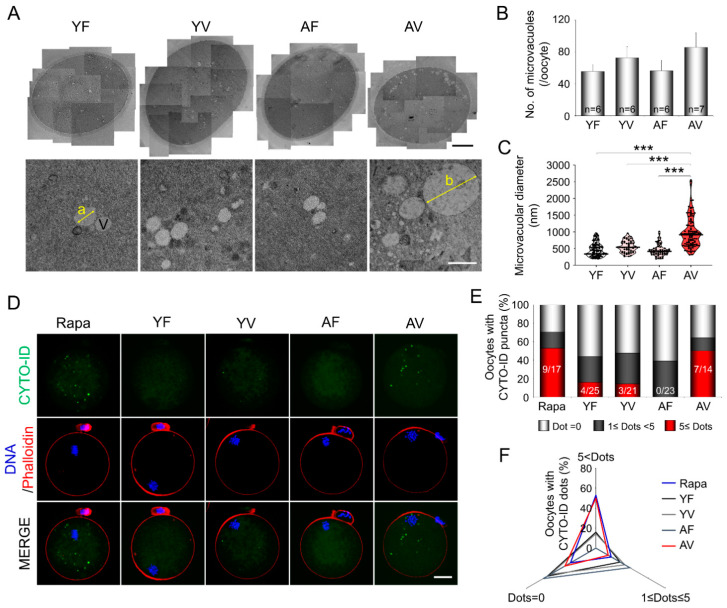
Ultrastructural analyses of fresh and vitrified/warmed (VW) oocytes from young and aged mice. (**A**) Transmission electron microscopy images showing the microvacuoles in fresh and VW oocytes (a = 722 nm, b = 1910 nm). Upper panel scale bar = 12 µm; Bottom panel scale bar = 1 µm. (**B**,**C**) Quantitative analyses for the diameter and number of microvacuoles in fresh and VW oocytes from young and aged mice (*n* = 6–7 oocytes in each group). Note that the size and number of microvacuoles increased in VW oocytes from aged mice. The horizontal black lines represent the median values of the microvacuolar diameter of oocytes. Statistical comparisons were performed using one-way ANOVA, followed by Tukey’s multiple comparisons test and Dunn’s multiple comparisons test. V: Microvacuole. *** *p* < 0.005. (**D**) Representative microscopic images of autophagosome visualized using CYTO-ID (Green) in oocytes from young and aged mice before or after vitrification. Confocal imaging analysis of the oocytes. (**E**,**F**) Quantitative analysis of CYTO-ID puncta in fresh and VW oocytes from young and aged mice (*n* = 14–25 oocytes in each group). YF: Young fresh, YV: Young vitrified, AF: Aged fresh, AV: Aged vitrified.

**Figure 5 cells-10-01563-f005:**
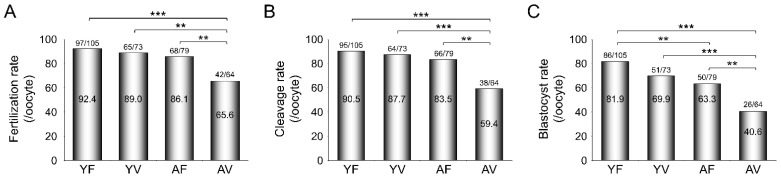
Effects of vitrification on the development of embryos from in vitro fertilization of oocytes from young and aged mice. (**A**–**C**) The rates of fertilization, cleavage, and blastocyst formation of oocytes from young (*n* = 10 mice, 1- to 2-month-old) and aged (*n* = 30 mice, 10- to 12-month-old) mice (*n* = 64–105 oocytes in each group) after in vitro fertilization. Statistical comparisons were performed using chi-squared test. ** *p* < 0.01, *** *p* < 0.005. YF: Young fresh, YV: Young vitrified, AF: Aged fresh, AV: Aged vitrified.

**Figure 6 cells-10-01563-f006:**
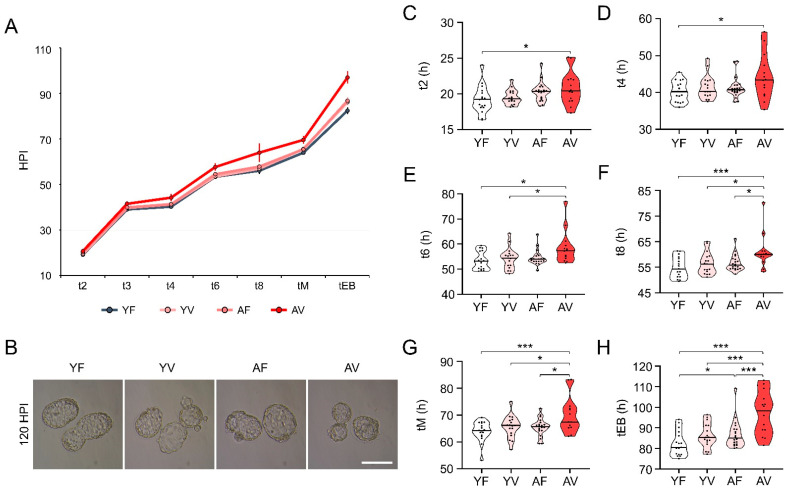
Comparison of embryo morphokinetics of fresh and vitrified/warmed (VW) oocytes from young and aged mice. (**A**–**H**) Comparison of morphokinetics of embryos developed from oocytes from young and aged mice after vitrification (*n* = 14–20 oocytes in each group). t2, t3, t4, t6, t8: time in hours post insemination (HPI) for the embryo to reach the 2-, 3-, 4-, 6-, 8-cell stage, respectively, tM: time for compaction, tEB: time for the embryo to start blastocoel formation. Statistical comparisons were performed using student’s *t*-test. Four independent experiments were performed with similar results. * *p* < 0.05, *** *p* < 0.005. Scale bar = 140 µm. YF: Young fresh, YV: Young vitrified, AF: Aged fresh, AV: Aged vitrified.

**Figure 7 cells-10-01563-f007:**
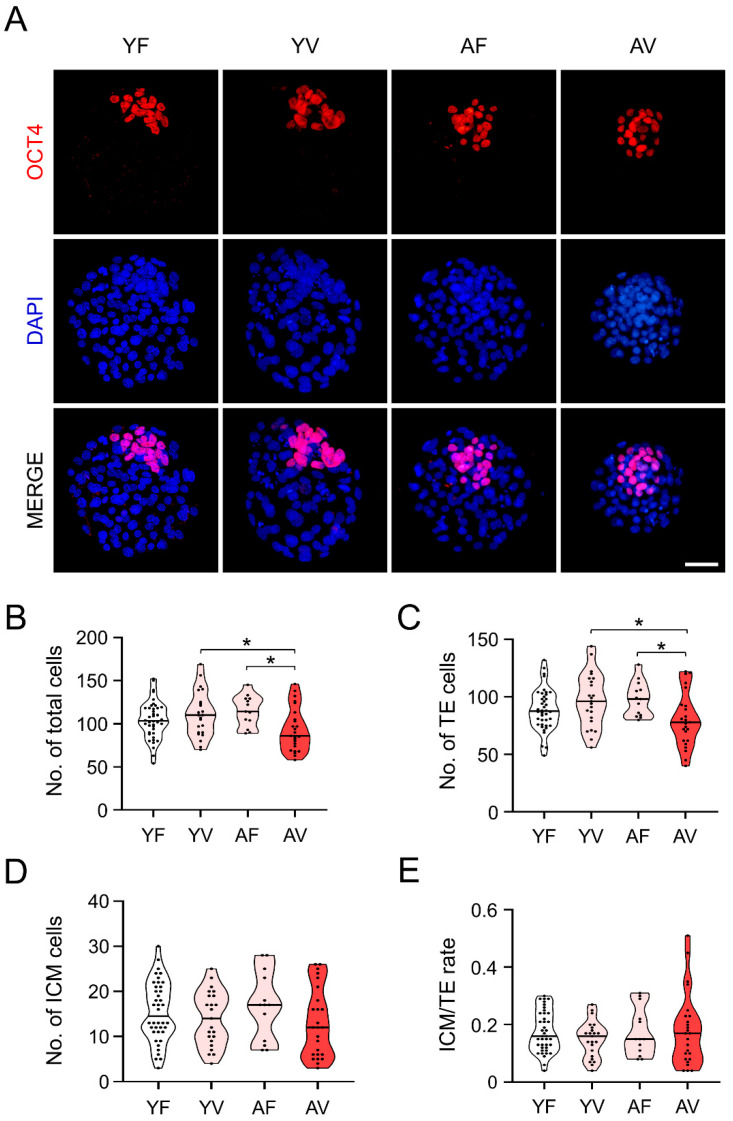
Evaluation of the effects of vitrification on blastocyst development in oocytes after in vitro fertilization. (**A**) Immunofluorescence staining for OCT4 [inner cell mass (ICM) marker], and DAPI (nuclei) in normal and abnormal blastocysts developed from fresh and vitrified/warmed oocytes from young and/or aged mice (*n* = 13–40 blastocysts in each group). (**B**–**E**) Quantitative analyses of the total cell number, trophectoderm (TE), ICM, and ICM/TE ratio in blastocysts from vitrified/warmed oocytes of young and aged mice. The total cell number was calculated by counting the DAPI-stained nuclei. The number of ICM and TE were counted from OCT4-positive and DAPI-positive cells without OCT4, respectively. The horizontal black lines represent the median values of total cell number, number of ICM and TE, and ICM/TE ratio. Statistical comparisons were performed using one-way ANOVA, followed by Tukey’s multiple comparisons test. Scale bar = 50 µm. * *p* < 0.05. YF: Young fresh, YV: Young vitrified, AF: Aged fresh, AV: Aged vitrified.

## Data Availability

Not applicable.

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
