# Peer review of "Advanced Maternal Age Deteriorates the Developmental Competence of Vitrified Oocytes in Mice"

_cells, 2021, doi:10.3390/cells10061563_

Round 1

Reviewer 1 Report

Lee et al. present a thorough evaluation of the interaction between maternal age and vitrification on the developmental competence of mouse oocytes.  The study is well designed and multiple endpoints are described.  I would only suggest some minor changes.

1) Line 106:  Please indicate if the amount of HSA used is based on volume (20% of total volume) or the ratio of weight to volume (20% w/v = 200 mg/ml).

2) Lines 120-133:  This section describes staining of oocytes, but Oct4 is included.  Results indicate that only blastocysts were stained for Oct4.  Please distinguish between protocols for staining oocytes and protocols for staining embryos.

3) Lines 174 and 184:  Two different times for IVF are provided - 6 h (line 174) and 8 h (line 184).  Were these supposed to be different for the different experiments?

Author Response

A point-by-point response to the reviewers

We deeply appreciate critical comments from the reviewers on our submitted manuscript. We think that all the comments from the reviewers are critical to clarify and strengthen the excellence of our resubmitted manuscript. We have tried our best to meet all the requests from the reviewer in the revised manuscript. We highlighted the modified/corrected words and/or sentences as yellow in the revised manuscript.

Reviewer 1

Lee et al. present a thorough evaluation of the interaction between maternal age and vitrification on the developmental competence of mouse oocytes. The study is well designed and multiple endpoints are described.  I would only suggest some minor changes.

1) Line 106: Please indicate if the amount of HSA used is based on volume (20% of total volume) or the ratio of weight to volume (20% w/v = 200 mg/ml).

--> As suggested by the reviewer, we added information for the HSA that is based on volume to volume (v/v) in the revised manuscript (line 107).

2) Lines 120-133: This section describes the staining of oocytes, but Oct4 is included.  Results indicate that only blastocysts were stained for Oct4. Please distinguish between protocols for staining oocytes and protocols for staining embryos.

--> We appreciate the reviewer’s critic of our mistake in the submitted manuscript. We separately described the protocols of immunofluorescence staining for oocytes and blastocysts as suggested by the reviewer in the revised manuscript (lines 136-141).

3) Lines 174 and 184:  Two different times for IVF are provided - 6 h (line 174) and 8 h (line 184).  Were these supposed to be different for the different experiments?

à We would like to show our appreciation for the reviewer’s detailed comment on the submitted manuscript. Although the methods in 2.8 and 2.9 were basically the same, we happened to describe the procedures differently. In section 2.9, MII oocytes were inseminated with sperms for 6 hours and washed for PN observation to separate zygotes from unfertilized oocytes. Approximately 8 h after insemination, fertilized zygotes were transferred to Primo Vision culture dish for time-lapse monitoring. We modified the sentence not to make the reviewer and readers confused with this matter in the revised manuscript (lines 193-195).

Reviewer 2 Report

The Authors, Lee and colleagues, tackled an important question, namely, how maternal age influences the outcome of oocyte vitrification, in the mouse model. There are only few basic studies in this area, in which the effect of maternal age on vitrification was taken into account. Overall, the study submitted is well conducted, well presented, timely and important.

I have the following remarks, which in my opinion warrant a revision.

Authors used oocytes of B6 inbred strain. This is good for standardization, but is not so good for generalisation (to other mice) or even extrapolation (to other species). How do the B6 results compare with those of other mouse studies in which different mouse strains were used e.g. PMID 21705053 (which the Authors have cited?). When it comes to extrapolating to other species, humans are not inbred. I recommend to add a statement that additional studies are needed in other mouse strains and animal species, in order to confirm the general validity of effects observed in this work.

How long were the vitrified oocytes stored in LN2 before thawing? Please provide this information in the manuscript.

The results shown in Figure 2A were analyzed using the chi-square test. Is this formally correct? Isn’t there a lower bound on the amount of data that is needed for chi-square test? I’m asking because one of your data is 0/22, and I may recall that the chi-square test is inappropriate when the frequencies are zero. Please check.

Regarding the ultrastructural analysis: the sections are ultra-thin, so, did the Authors examine only some sections per oocyte, or all of them? I suggest to be very cautios, because unless you examine all sections, the detection (or lack) of microvacuoles may be just by chance. Please provide this information in the manuscript, on how representative the sections are of the whole, and how you counted the sections. 

In Figure 5C, do the differences e.g. the lower blastocyst rate of the AV group persist even if you normalize to the 2-cell stage?  At line 176 the Authors wrote: ”blastocyst formation (number of blastocysts/2-cell embryos)”, whereas the Y axis of Fig.5C shows blastocysts / oocyte. Please clarify. If you take the 2-cell rate as rate of fertilization (which is fine by me), then you might want to consider that some of the thawed oocytes might have undergone spontaneous activation. Please clarify.

Since the Authors showed that the differences between the old and young oocytes become more prominent during the course of preimplatation development (Figure 6), what do you think would happen after embryo transfer to uterus? As an example,  doi: https://doi.org/10.1101/410514 could be an interesting reading. Please comment on this.

The take-home message of the study is important: “AMA further aggravates the vulnerability of VW oocytes to cryo-damage”. I suggest the Authors mine available gene expression datasets (transcriptome, proteome) to see what the major differences between young and old oocytes are, before these are vitrified. This might enrich the Authors’ results with a mechanistic dimension e.g. ‘cryoprotective’ gene products are depleted in aged oocytes. Just an idea. As possible examples of relevant studies: PMID 24686459, PMID 15317747, PMID 32951297, PMID 16905767 (there are more than just these four).

Minor remarks

Were the sperm donors used for IVF also B6? Please add this information.

Was the KSOM medium supplemented with aminoacids? Please add this information.

Why are the mitochondria of Figure 3A green, if you used MitoTracker Red CMXRos ? Please clarify.

Thank you for the opportunity to read your interesting study.

Author Response

A point-by-point response to the reviewers

We deeply appreciate critical comments from the reviewers on our submitted manuscript. We think that all the comments from the reviewers are critical to clarify and strengthen the excellence of our resubmitted manuscript. We have tried our best to meet all the requests from the reviewer in the revised manuscript. We highlighted the modified/corrected words and/or sentences as yellow in the revised manuscript.

Reviewer 2

The Authors, Lee and colleagues, tackled an important question, namely, how maternal age influences the outcome of oocyte vitrification, in the mouse model. There are only a few basic studies in this area, in which the effect of maternal age on vitrification was taken into account. Overall, the study submitted is well conducted, well presented, timely, and important.

I have the following remarks, which in my opinion warrant a revision.

Authors used oocytes of B6 inbred strain. This is good for standardization, but is not so good for generalisation (to other mice) or even extrapolation (to other species). How do the B6 results compare with those of other mouse studies in which different mouse strains were used e.g. PMID 21705053 (which the Authors have cited?). When it comes to extrapolating to other species, humans are not inbred. I recommend to add a statement that additional studies are needed in other mouse strains and animal species, in order to confirm the general validity of effects observed in this work.

--> We are thankful for a constructive comment from the reviewer. As suggested by the reviewer, we initially performed all the experiments in NOT ONLY BL6 BUT ALSO BDF1 AND ICR strains. After we analyzed the initial data of three different mouse strains, we validated that the BL6 strain is the most suitable strain to pursue what we have done for the effects of AMA on oocyte vitrification. As the reviewer mentioned, humans are, so to speak, outbred although BL6 is an inbred strain. However, humans are very vulnerable with respect to reproduction and we think that the reproductive potential of BL6 is similar to that of humans. Even if we think that the BL6 mouse strain is the right strain to study the effects of AMA on oocyte vitrification, we absolutely agree with the reviewer on this matter. Thus, we included the above sentence suggested by the reviewer in the revised manuscript (lines 443-445).

How long were the vitrified oocytes stored in LN2 before thawing? Please provide this information in the manuscript.

--> Many references suggest that the storage duration does not significantly affect the outcomes of vitrified/warmed oocytes and embryos. Thus, in our experiments, we stored vitrified oocytes for 10 min and warmed them for the experiments. As suggested by the reviewer, we described the storage duration of vitrified oocytes in the revised manuscript (lines 111-112).

The results shown in Figure 2A were analyzed using the chi-square test. Is this formally correct? Isn’t there a lower bound on the amount of data that is needed for chi-square test? I’m asking because one of your data is 0/22, and I may recall that the chi-square test is inappropriate when the frequencies are zero. Please check.

--> We deeply appreciate the reviewer’s critical comment on the statistical analyses that we performed. We thought that chi-square test is appropriate and did not pay attention to the statistical issue with zero frequency in the data. During the revision process, we found out that it is appropriate to use Fisher’s exact test instead of Chi-square test in this case as suggested by the reviewer. Accordingly, we analyzed the data with Fisher’s exact test and described the statistical analyses accordingly in the revised manuscript (lines 204-205, 256-257, and 275-276).

Regarding the ultrastructural analysis: the sections are ultra-thin, so, did the Authors examine only some sections per oocyte, or all of them? I suggest to be very cautious, because unless you examine all sections, the detection (or lack) of microvacuoles may be just by chance. Please provide this information in the manuscript, on how representative the sections are of the whole, and how you counted the sections.

--> As we all know, ultrastructural analysis using TEM images is mainly qualitatively examined, but not quantitatively. Although we were not able to include the number of TEM images as many as other experiments, we tried our best to be objective without any prejudice to evaluate the quantity with respect to the number and size of microvacuoles in the oocyte. We looked at several semi-thin sections of each oocyte (at least 6 to 7 independent oocytes were examined in each group) and picked the most representative section among them for each oocyte for the microvacuole counting by two independent observers. The diameter of microvacuoles was similarly examined in ultrathin sections of the oocytes. According to the reviewer’s comment, we added detailed procedures for the analyses of the number and diameter of microvacuoles in the oocytes in the revised manuscript (lines 162-164).

In Figure 5C, do the differences e.g. the lower blastocyst rate of the AV group persist even if you normalize to the 2-cell stage? At line 176 the Authors wrote: ”blastocyst formation (number of blastocysts/2-cell embryos)”, whereas the Y axis of Fig.5C shows blastocysts / oocyte. Please clarify.

--> We would like to show our sincere appreciation for the reviewer’s thorough review. The reviewer found a serious mistake that we did not find in our submitted manuscript. It was the wrong description. In section 2.8, the rates of fertilization, cleavage, and blastocyst formation were all analyzed with survived oocytes but not embryos. As suggested by the reviewer, we corrected them in the revised manuscript (lines 186-187).

If you take the 2-cell rate as the rate of fertilization (which is fine by me), then you might want to consider that some of the thawed oocytes might have undergone spontaneous activation. Please clarify.

--> We absolutely agree with the reviewer on this issue. If we use the number of 2-cell embryos for the fertilization rate, some parthenogenic embryos without fertilization could be included in the data. Although we consider the number of 2-cell embryo for fertilization rate in Figure 5A, we also validated the fertilization of vitrified/warmed oocytes by observing 2PN formation for time-lapse monitoring in Figure 6. We barely found parthenogenic embryos in leftover dishes so that we decided to use 2-cell count as fertilization rate as in many published articles.

Since the Authors showed that the differences between the old and young oocytes become more prominent during the course of preimplantation development (Figure 6), what do you think would happen after embryo transfer to uterus? As an example, doi: https://doi.org/10.1101/410514 could be interesting reading. Please comment on this.

--> We absolutely agree with the reviewer on this issue. We should have done embryo transfers to make sure that the blastocysts from the AV group have lower implantation potentials and pregnancy outcomes. Although we initially planned to perform embryo transfer, as all we know, it is extremely difficult to have enough numbers of blastocysts from AV groups for embryo transfer. Although we were not able to do the embryo transfer in this study, we would like to include a sentence that mentions this issue in the discussion of the revised manuscript as suggested by the reviewer (lines 430-431).

The take-home message of the study is important: “AMA further aggravates the vulnerability of VW oocytes to cryo-damage”. I suggest the Authors mine available gene expression datasets (transcriptome, proteome) to see what the major differences between young and old oocytes are before these are vitrified. This might enrich the Authors’ results with a mechanistic dimension e.g. ‘cryoprotective’ gene products are depleted in aged oocytes. Just an idea. As possible examples of relevant studies: PMID 24686459, PMID 15317747, PMID 32951297, PMID 16905767 (there are more than just these four).

--> We really appreciate the thoughtful comment of the reviewer. As suggested by the reviewer, we also wanted to find a mechanistic link between the phenotypes of oocytes from AV groups and gene expression from the references. Whereas we mined gene expression datasets published before and selected several genes, including to perform RT-PCR between groups in this study, we were not able to find a significant difference in gene expression in oocytes among groups.

However, we absolutely agree with the reviewer on the fact that it is critical to mention the links between the phenotypes and gene expression profiles between young and aged, and/or fresh and vitrified oocytes. Thus, we included a sentence to show possible links between phenotypes and gene expression with references in the discussion section in the revised manuscript (lines 438-440).

Minor remarks

Were the sperm donors used for IVF also B6? Please add this information.

--> We used BDF1 male mice as sperm donors in this study. We mentioned this in the revised manuscript (lines 180-181).

Was the KSOM medium supplemented with amino acids? Please add this information.

--> Yes. We used KSOM medium supplemented with amino acids. We added the information accordingly as suggested by the reviewer in the revised manuscript (lines 97-99).

Why are the mitochondria of Figure 3A green, if you used MitoTracker Red CMXRos ? Please clarify.

--> We are deeply sorry to make the reviewer confused with it. The reviewer’s comment is absolutely right. We used MitoTracker Red to visualize mitochondria as red in oocytes, but red fluorescence was not as bright as green in oocytes. Thus, we converted “Red” fluorescence to “Green as pseudo-color” using the software. We described the conversion process of fluorescence color in the revised manuscript accordingly (lines 148-151)